# IL-2K35C-moFA, a Long-Acting Engineered Cytokine with Decreased Interleukin 2 Receptor α Binding, Improved the Cellular Selectivity Profile and Antitumor Efficacy in a Mouse Tumor Model

**DOI:** 10.3390/cancers14194742

**Published:** 2022-09-28

**Authors:** Xiaoze Wang, Gang Chen, Lei Nie, Zhenhua Wu, Xinzeng Wang, Chenxiao Pan, Xuchen Chen, Xiaobei Zhao, Jie Zhu, Qiaojun He, Haibin Wang

**Affiliations:** 1College of Pharmaceutical Sciences, Zhejiang University, Hangzhou 310058, China; 2Zhejiang Bioray Biopharmaceutical Co., Ltd., Taizhou 318000, China; 3Bioray Pharmaceutical Inc., San Diego, CA 92121, USA

**Keywords:** interleukin 2, cancer immunotherapy, fatty acid conjugation, long half-life

## Abstract

**Simple Summary:**

The application of IL-2 for treating cancer is limited owing to its toxicity and short half-life. Its high binding ability to IL-2 receptor α expands immunosuppressive Treg cells, which represents an undesirable toxicity in cancer immunotherapy. Moreover, its small molecular size is the reason for its short half-life. We solved these problems by using a covalent modification strategy of IL-2 variant IL-2K35C with fatty acid by maleimide chemistry, namely, IL-2K35C-moFA. The experiments performed in vitro and in vivo proved that IL-2K35C-moFA is a novel immunotherapeutic agent with the potential to selectively stimulate CD8^+^ T cells and NK cells. Compared to IL-2WT, IL-2K35C-moFA showed a specifically reduced potency for the stimulation of Treg cells. Our results also showed that fatty acid conjugation appears to be effective in half-life extension. The combination of selective lymphocyte expansion and its long half-life means IL-2K35C-moFA should be evaluated as a potential human immunotherapeutic in the future.

**Abstract:**

Human interleukin 2 (IL-2) has shown impressive results as a therapeutic agent for cancer. However, IL-2-based cancer therapy is limited by strong Treg amplification owing to its high binding affinity to IL-2 receptor α (IL-2Rα) and its short half-life owing to its small molecular size. In this study, we solved these problems using a covalent modification strategy of the IL-2 variant, i.e., substituting cysteine (C) for lysine (K) at position 35, using octadecanedicarboxylic acid through maleimide chemistry, creating IL-2K35C-moFA. IL-2K35C-moFA was equipotent to human IL-2 wild type (IL-2WT) in activating tumor-killing CD8^+^ memory effector T cells (CD8^+^ T) and NK cells bearing the intermediate affinity IL-2 receptors, and less potent than IL-2WT on CTLL-2 cells bearing the high-affinity IL-2 receptors. Moreover, it was shown to support the preferential activation of IL-2 receptor β (IL-2Rβ) over IL-2Rα because of the mutation and fatty acid conjugation. In a B16F10 murine tumor model, IL-2K35C-moFA showed efficacy as a single dose and provided durable immunity for 1 week. Our results support the further evaluation of IL-2K35C-moFA as a novel cancer immunotherapy.

## 1. Introduction

Interleukin-2 (IL-2) is a well-studied cytokine that regulates several key functions of the immune system. Human IL-2 was first identified in the early 1980s, and it acts as a variably glycosylated 15.5 kilodalton (kDa) protein [1,2,3,4]. IL-2, known as “T cell growth factor”, was discovered in 1976 [5]. Recombinant human IL-2 (Proleukin, aldesleukin) was approved by the United States Food and Drug Administration for metastatic melanoma and renal cell cancer [6]; however, adverse events associated with high-dose IL-2, including capillary leak syndrome (VLS), limit its therapeutic use [7,8,9].

IL-2 exerts stimulatory and regulatory functions by binding to monomeric, dimeric, and trimeric receptors [10,11]. IL-2 binds to monomeric receptor IL-2Rα with a weak affinity (Kd: ∼10^−8^ M), dimeric receptors composed of IL-2Rβ and γc subunits with an intermediate affinity (Kd: ∼10^−9^ M), and trimeric receptors composed of IL-2Rα, IL-2Rβ, and γc subunits with a strong affinity (Kd: ∼10^−9^ M) [12,13]. 

The binding of IL-2 to IL-2Rβγc leads to the desired expansion of CD8^+^ T cells at high doses [14,15]. IL-2 also binds to IL-2Rαβγc, expanding immunosuppressive CD4^+^CD25^+^Foxp3^+^ Tregs, which express high constitutive levels of IL-2Rα [16]. The expansion of Tregs represents an undesirable stimulation of IL-2 in cancer immunotherapy [12,14,17,18]. 

Human serum albumin (HSA), the most abundant protein in the circulatory system, plays a key role in drug transport and metabolism [19,20,21,22]. Fatty acids (FAs) can bind to human HSA, making them a potent method to extend the half-lives of drugs with poor stability [23,24,25,26]. 

In our study, we developed a simple, economical method to generate an IL-2 molecule, which can not only biasedly activate the antitumor activity of immune cells, but also has a longer serum half-life compared with IL-2WT. We generated several cysteine mutants of IL-2 that exhibit lower IL-2Rα binding than IL-2WT. The conjugation to FA further lowers IL-2Rα binding due to the steric hindrance of the FA molecule. The binding of HSA by FA in the serum during the circulation could reduce IL-2Rα binding to an even greater extent, which will reduce the activation of Treg cells and other cells that overexpress IL-2Rα in vivo. Among those cysteine mutants, IL-2K35C-moFA showed a significant decrease in IL-2Rα binding ability, while maintaining IL-2Rβ binding capacity both in vitro and in vivo. IL-2K35C-moFA also exhibited a longer half-life and more durable antitumor efficacy than IL-2WT.

## 2. Materials and Methods

### 2.1. Construction and Purification of IL-2 Muteins

The overlap polymerase chain reaction was used to generate the Fc-tagged (human IgG1) IL-2 mutants (IL-2K35C, IL-2K43C, IL-2K64C) in which the thrombin cleavage sequence LVPRGS was inserted between the Fc tag and IL-2 mutant. Among the three mutants, only Fc-IL-2K35C showed satisfactory expression. The primer sequences of IL-2K35C are shown in Table 1. The Fc-IL-2K35C fusion protein was purified by protein A affinity and was then digested with GST-tagged thrombin. Protein A and glutathione beads were used to remove the Fc tag and GST-tagged thrombin, respectively.

### 2.2. FA Conjugation

Maleimide-modified octadecanedicarboxylic acid (moFA) was synthesized by a chemical synthesis laboratory in Zhejiang Bioray Biopharmaceutical Co., Ltd. Before conjugation, IL-2K35C was reduced by 6 molar equivalents of tris (2-carboxyethyl) phosphine (TCEP) (51805-45-9, Merck Millipore, UK) for 2 h in a reaction buffer containing 35 mM sodium citrate, 2 mM EDTA, and 154 mM NaCl; the pH was 5. The reduced IL-2 was mixed with 2 molar equivalents of moFA in reaction buffer containing 5% DMA for 2 h for conjugation at 18 °C.

### 2.3. RP-HPLC

The conjugated product was analyzed by reversed-phase HPLC (RP-HPLC) using an Agilent 1200 HPLC system equipped with a reversed-phase C8 column (3.5 μm, XBridge® C8). The volume of the loaded sample was 20 μL. The mobile phase consisted of 0.1% trifluoroacetate in water (A) and 0.1% trifluoroacetate in acetonitrile (B), with an initial 35% B for 5 min, which was increased to 80% B in 30 min at a flow rate of 1 ml/min.

### 2.4. Mass Analysis by LC-MS 

Protein mass analysis was performed on Waters ACQUITY UPLC with a Waters Xevo-G2S Q-TOF mass spectrometer (LC-MS, Waters, Milford, MA, USA). The following parameters were used: capillary voltage 2.5 kV, sampling voltage 100 V, source temperature 120 °C, desolvation temperature 500 °C, and desolvation gas flow 800 L/h. The *m*/*z* from 250 to 4000 was obtained. Mobile phase: (A) 0.1% formic acid in water; (B) 0.1% formic acid in acetonitrile. Gradient: 10% B to 90% B in 5 min. Flow rate was set to 0.2 mL/min. The data were processed using BiopharmaLynx software (1.3, Waters, Milford, MA, USA).

### 2.5. Mapping Analysis by LC-MS/MS

Disulfide linkage was analyzed at the peptide level by LC-MS/MS. The analysis was performed on a Waters ACQUITY UPLC with a Waters Xevo-G2S Q-TOF mass spectrometer (LC-MS/MS, Waters, Milford, MA, USA). Waters BEH 300 C18 column (2.1 × 150 mm, 1.7 μm) was equipped. The mobile phase consisted of 0.1% formic acid in water (A) and 0.1% formic acid in acetonitrile (B); 10 µL of the digested sample was loaded with an initial 2% B for 3 min and increased to 50% B over 47 min at 40 °C. Flow rate was set to 0.2 mL/min. The data were processed using the BiopharmaLynx 1.3.3.

### 2.6. Indirect Enzyme-Linked Immunosorbent Assay (ELISA)

Polystyrene 96-well microtiter plates (PolySorp Nunc, Chicago, IL, USA) were coated with an IL-2Rα Fc tag (ILA-H5251, ACROBiosystems, Beijing, China) at a concentration of 2 μg/mL at 4 °C overnight, blocked with 5% BSA in PBS, and then IL-2WT (H4113, ACROBiosystems, China), IL-2K35C, and IL-2K35C-moFA were added to 100 μL of 1:1 serial dilution in 1% BSA-PBST. The dilutions ranged from 20 to 0.039 ng/ml. After washing, biotin-labeled anti-human IL-2 (517605, BioLegend, San Digo, CA, USA) was added and incubated for 1 h at 37 °C. Thereafter, 100 μL/well HRP-streptavidin (ab7403, Abcam, Cambridge, MA, USA) diluted 1:5000 was added, followed by adding 100 μL tetramethylbenzidine substrate (TMB-S-001, Huzhou InnoReagents Co., Ltd., Huzhou, China). Then, 50 μL 2N H_2_SO_4_ was used to stop the reaction. The mean optical density at 450 nm (OD450) was determined by Spectramax M5 (Molecular Devices, San Jose, CA, USA).

To detect the binding of samples to IL-2Rβ, IL-2Rβ-mFc (10696-H05H, Sino Biological, Beijing, China) was coated at a concentration of 2 μg/mL at 4 °C overnight. IL-2WT, IL-2K35C, and IL-2K35C-moFA were added at concentrations of 20–0.01 ng/mL. For other steps, please refer to the ELISA method for IL-2Rα.

### 2.7. CTLL-2 Cell Proliferation Assay

CTLL-2 (ATCC) cells at a density of 1 × 10^5^ cells/mL were seeded in a 96-well-plate at 50 μL/well. IL-2WT, IL-2K35C, and IL-2K35C-moFA solutions were prepared at concentrations ranging from 50 to 0.002 ng/mL. Fifty microliters of each diluted sample was added to each well. The plate was incubated at 37 °C and 5% CO_2_ for 48 h. Next, 100 μL CellTiter-Glo® (G7572, Promega Corporation, Madison, WI, USA) was added; luminescence was recorded suing a Spectramax M5.

### 2.8. p-STAT5 Assay in Treg and CD8^+^ T Cell

Tregs (PB009-4F-2, Allcells, Shanghai, China) at a density of 1 × 10^6^ cells/mL were seeded in InstantOne ELISA kit (85-86112-11, Invitrogen, Carlsbad, CA, USA) microplates at 20 μL/well. The plates were then incubated for 2 h at 37 °C. IL-2WT, IL-2K35C, and IL-2K35C-moFA were added to each well at the concentration of 0.1 μg/mL and 0.01 μg/mL. Ten microliters of cell lysis mix (5× was added to each of the InstantOne ELISA assay wells after shaking (~300 rpm) for 10 min. The absorbance of the plates was measured at 450 nm using a Spectramax M5. 

CD8^+^ T cells (PB009-3-C, Allcells, Shanghai, China) at a density of 1 × 10^6^ cells/mL were seeded into an InstantOne phospho-STAT5 ELISA kit microplate at 20 μL/well and incubated for 2 h at 37 °C. IL-2WT, IL-2K35C, and IL-2K35C-moFA were added to each well at a concentration of 2 μg/mL. For other steps, please refer to the Tregs method.

### 2.9. Animal Experiments

Female C57BL/6J mice aged 6 weeks (purchased from the Vital River Laboratory Animal Technology Co. Ltd., Beijing, China) were used for animal studies. Animals were cared for in accordance with the Guide for the Care and Use of Laboratory Animals in China. 

### 2.10. Half-Life Analysis in Mice

Mice were randomly divided into IL-2K35C-moFA and IL-2WT groups (*n* = 5 per group). IL-2K35C-moFA or IL-2WT was injected intravenously into mice at the same concentration (200 μL, 0.1 mg/kg), and the half-lives of IL-2K35C-moFA and IL-2WT were calculated from concentrations measured from blood samples collected over the course of 72 h. A sensitive sandwich ELISA was established to estimate serum IL-2WT and IL-2K35C-moFA concentrations. The plots of serum concentration against time were fitted suing the Rosenblueth method of the noncompartment model. 

### 2.11. Immune Cell Phenotyping in Mice

Three groups of C57BL/6J mice (*n* = 6 per group) received subcutaneous injection once daily with PBS, 0.5 mg/kg IL-2WT for 5 days or twice a week with 0.5 mg/kg IL-2K35C-moFA. Two hours after the last injection on day 7, the mice were euthanized. Spleen single-cell suspensions were obtained by grinding of spleens, followed by passing through 100 mm sterile cell strainers. Cells were stained with fluorescent-labeled antibodies against markers indicative of Tregs, NK, and CD8^+^ T cells. The following antibodies were used: FITC anti-mouse CD3 antibody [17A2] (E-AB-F1013C, Elabscience, Houston, TX, USA), PerCP/Cyanine5.5 anti-mouse CD4 antibody [GK1.5] (E-AB-F1097UJ, Elabscience, Houston, TX, USA), PE rat anti-mouse CD8a antibody [53-6.7] (553032, BD Biosciences, USA), APC anti-mouse CD161/NK1.1 antibody [PK136] (E-AB-F0987UE, Elabscience, Houston, TX, USA), APC rat anti-mouse CD25 [PC-61.5.3] (E-AB-F1102E, Elabsicence, Houston, TX, USA), PE anti-mouse Foxp3 antibody [3G3] (E-AB-F1238D, Elabscience, Houston, TX, USA). PE mouse IgG1, κ isotype control [MOPC-21] (E-AB-F09792D, Elabscience, USA), and APC mouse IgG1 k isotype control (BD Biosciences, USA) were used for negative controls. The Tregs were gated from the CD4^+^ T cells as the CD25^+^Foxp3^+^ population. The NK cells were identified as the CD3^−^NK1.1^+^ population. The CD8^+^ T cells were gated as the CD3^+^CD8^+^ T population. Samples were acquired and analyzed using BD Accuri C6 and quantified by FlowJo software (v.X.0.7).

### 2.12. B16F10 Murine Tumor Model

The viability of B16F10 melanoma cells was assessed by trypan blue staining. The cell batch was considered for injection only when cell viability was >95%. C57BL/6J mice were implanted subcutaneously into the right flank with B16F10 cells (5 × 10^5^ cells per animal). When palpable tumors were established, mice were subcutaneously administered control PBS, IL-2WT (3 mg/kg), or IL-2K35C-moFA (3 mg/kg). The tumor size and body weight were monitored on day 7.

### 2.13. Data Analysis

The GraphPad Prism 8.0 software (GraphPad Software, San Diego, CA, USA) was used for statistical analyses. Statistical significances of differences were determined using the two-tailed unpaired Student t-test.

## 3. Results

### 3.1. Screen/Design IL-2 Mutants 

The IL-2Rα subunit forms the largest of the three IL-2/IL-2 receptor interfaces. [27] The nature of the IL-2/IL-2Rα interface revealed a striking dichotomy of hydrophobic centers and polar peripheries, featuring ion bonds dominated by IL-2 residues K35, T37, R38, T41, F42, K43, F44, Y45, E61, E62, K64, P65, L72, and Y107 (Figure 1A). Three essential residues, K35, K43, and K64, were individually mutated to cysteine (K35C, K43C, and K64C, respectively) for conjugation with FA (Figure 1A). These mutants are designed as fusion proteins with Fc tags, along with regions that provide thrombin cleavage sites, and are expressed in CHO cells.

### 3.2. Cloning, Expression, and Purification of IL-2K35C

Among the three mutants, only IL-2K35C was well expressed in CHO cells, with a protein yield of 25 mg/L. The protein sequence of Fc-IL-2K35C is shown in Figure 1B. 

The Fc-tagged IL-2K35C protein has a theoretical molecular weight of 42 kDa. The protein was purified by protein A affinity chromatography and size exclusion chromatography. SDS-polyacrylamide gel electrophoresis (SDS-PAGE) was performed to demonstrate the fusion protein message. As shown in Figure 1C, Fc-tagged IL-2K35C migrated as a single prominent band with an apparent molecular mass of 42 kDa. Under nonreduced conditions, the protein appeared to be a dimer, and under reduced conditions (1 mM DTT), the protein was a monomer. After Fc tag was cleaved by GST-tagged thrombin, IL-2K35C showed a prominent band at 15.5 kDa under reduced conditions (Figure 1D).

### 3.3. moFA Conjugation and Identification by LC-MS

The moFA and IL-2K35C showed well-defined peak groups in RP-HPLC. The moFA peak was eluted at 13.564 min, and the IL-2K35C protein was eluted at 23.501 min (Figure 2A). After conjugation, IL-2K35C-moFA eluted at 24.893 min, and there was a small amount of uncoupled impurities (Figure 2A). Conjugate masses were identified using liquid chromatography-mass spectrometry (LC-MS). As shown in Figure 2B, before conjugation, the protein mass of IL-2K35C was 16,104.94 Da. After conjugation, the protein mass of the main product was 16,960.48 Da, an increase of 855.54 Da (Figure 2C). The mass of IL-2K35C-moFA increased by the molecular weight of one FA. Next, peptide mapping was performed to analyze the disulfide bond positions and conjugation sites. As shown in Figure 2D, peptides HLQCLEEELKPLEEVLNLAQSK and QSETTFMCEYADETATIVEFLNR formed the correct intramolecular disulfide bond, suggesting that the conjugation protocol has no effect on the intramolecular disulfide bond. Figure 2E confirms that the conjugation site is the mutated cysteine at site 35. The above results demonstrate that only one FA molecule is site specifically conjugated to IL-2 mutant at position 35.

### 3.4. Binding Affinity to Receptors by ELISA

ELISA was used to measure the binding affinities of IL-2K35C and IL-2K35C-moFA to IL-2Rα and IL-2Rβ. As expected, the recognition of IL-2K35C and IL-2K35C-moFA to IL-2Rα decreased significantly compared with that of IL-2WT (Figure 3A); there was a minimum amount of IL-2Rα binding by IL-2K35C-moFA. The binding ability of IL-2Rβ was also evaluated. Among the three—IL-2WT, IL-2K35C, and IL-2K35C-moFA—IL-2K35C-moFA showed a similar binding ability to IL-2WT, which suggests the conjugation at residual 35 does not cause the activity change. IL-2K35C showed a higher binding ability to IL-2WT (Figure 3B). We believe the difference is due to the higher local concentration of dimeric IL-2K35C (Figure 1C), whereas IL-2WT and IL-2K35C-moFA are monomers. 

### 3.5. Effects on IL-2-Dependent Proliferation of CTLL-2 Cells

The cytotoxic T cell line CTLL-2 proliferates in response to IL-2, which requires stimulation by IL-2 through IL-2Rαβγc, making it suitable for the study of IL-2-dependent proliferation and signaling. For determining whether IL-2K35C and IL-2K35C-moFA influence IL-2-dependent proliferation of CTLL-2 cells, 50 µL of 1:1 serial dilution was added to the culture medium. The dilutions ranged from 50 to 0.002 ng/mL. As shown in Figure 3C, IL-2K35C and IL-2K35C-moFA largely inhibit IL-2-dependent CTLL-2 proliferation. For IL-2WT, the EC50 was 4.889 ng/mL, but it increased to 13.99 ng/mL in the IL-2K35C group and to 26.17 ng/mL in the IL-2K35C-moFA group. The EC50 of IL-2K35C-moFA was higher than that of IL-2K35C, indicating that the steric hindrance of FA decreased the binding ability to IL-2Rα and resulted in an ineffective induction of CTLL-2 cells.

### 3.6. Effects on IL-2-Dependent p-STAT5 of Treg and CD8^+^ T Cells

Regulatory T (Treg) cells express abundant amounts of all three IL-2 receptors. To definitively establish the role of IL-2K35C-moFA in Tregs in vitro, this study evaluated the downstream signaling pathway marked by p-STAT5 using an ELISA kit. As shown in Figure 3D, p-STAT5 was stimulated by IL-2 in a dose-dependent manner. Figure 3E shows that IL-2K35C and IL-2K35C-moFA exhibited decreased stimulation of Treg cells at different concentrations (0.01 μg/mL and 0.1 μg/mL). Moreover, the activation ability of IL-2K35C-moFA was lower than that of IL-2K35C. In contrast to Tregs, CD8^+^ T cells express abundant amounts of IL-2Rβ and γc. To test whether our strategy influenced CD8^+^ T effector cell function, we assessed p-STAT5 activity in CD8^+^ T cells. As shown in Figure 3F, there were no significant differences in p-STAT5 activity among the IL-2WT-, IL-2K35C-, and IL-2K35C-moFA-treated groups, indicating that IL-2K35C-moFA maintained CD8^+^ T cell activation, while PEGlyated IL-2s, such as THOR-707 [28,29], exhibited a fold lower activity compared with IL-2WT.

### 3.7. IL-2K35C-moFA Demonstrates Increased Half-Life in Mice

We next determined the half-lives of IL-2K35C-moFA and IL-2WT in C57BL/6J mice. A total of 2 μg of each protein was injected intravenously. The time points for the blood collection were at 5 min, 15 min, 30 min, 1 h, 2 h, 4 h, 8 h, 24 h, 48 h, and 72 h post administration. The collected blood samples were centrifuged at 3000 rpm for 10 min to separate the blood plasma. For comparison, the first value in the serum at 5 min was set to 100%. As shown in Figure 4A, IL-2WT exhibited a rapid decrease in serum concentrations within 2 h. In contrast, IL-2K35C-moFA sustained serum concentration levels for about 24 h. We observed a striking 8.42-fold increase in half-life from 1.239 h for IL-2WT to 10.431 h for IL-2K35C-moFA.

### 3.8. IL-2K35C-moFA Drives Expansion of CD8^+^ T and NK Cells in Mice

Splenocytes collected from wild-type mice were analyzed using multiparametric flow cytometry measuring cell-specific proliferation. The gating strategy for each cell type examined is visually represented in Appendix A. As shown in Figure 4B, treatment with IL-2WT resulted in a 5.9-fold increase in Tregs expansion on average, compared with the PBS-treated controls, while treatment with IL-2K35C-moFA resulted in an approximate 2.1-fold expansion of Tregs, which was due to the normal IL-2Rβγc signaling in Tregs. In contrast to the greater expansion of Tregs by IL-2WT, IL-2K35C-moFA preferentially expanded CD8^+^ T and NK cell populations (Figure 4C,D). IL-2K35C-moFA treatment induced a 5.3-fold expansion of CD8^+^ T cells (Figure 4C), and 6.1-fold expansion of NK cells (Figure 4D). Treatment with IL-2WT induced a comparable expansion of CD8^+^ T (4.5-fold) and NK cells (5.6-fold) to that of the IL-2K35C-moFA-treated group (Figure 4C,D). The splenic immune cell profiles and p values of the three groups are shown in Appendix A. These results suggest that IL-2K35C-moFA drives the expected proliferation of CD8^+^ T and NK cells, without driving a great expansion of Tregs in mice.

### 3.9. Therapeutic Efficacy and Safety in B16F10 Tumor Model

The therapeutic efficacy of IL-2K35C-moFA was tested in tumor-bearing C57BL/6J mice. When the tumor volume reached 200 mm^3^, mice received a single dose of 3 mg/kg IL-2K35C-moFA. The control groups received IL-2WT (3 mg/kg) or PBS twice per day for four consecutive days (Figure 4E). Tumor growth inhibition (TGI) and body weight were measured twice per week. As shown in Figure 4F, the body weight of the IL-2K35C-moFA group was comparable to that of the PBS group, indicating that IL-2K35C-moFA was nontoxic. Body weight significantly decreased on day 7 in the IL-2WT group, indicating toxicity. The results showed that IL-2K35C-moFA reduced toxicity to a greater extent than IL-2WT.

Both IL-2WT and IL-2K35C-moFA apparently inhibited tumor growth. IL-2K35C-moFA maintained its tumor-inhibiting effect for 7 days after only one injection, indicating that the coupled FA molecule significantly prolonged the half-life (Figure 4G). IL-2WT appears to have a stronger tumor-suppressive effect than IL-2K35C-moFA (Figure 4G), possibly because IL-2K35C-moFA has fewer IL-2 molecules at the same dose. Therefore, the optimal dosage of IL-2K35C-moFA for TGI must be optimized.

## 4. Discussion

This report describes the development of IL-2K35C-moFA and its potential as an immunotherapeutic agent that selectively activates intermediate affinity IL-2Rβγc for cancer therapy. This strategy is a novel IL-2 drug design for biologic therapy using FA to alter the immunologic, pharmacokinetic, and biodistribution profile of IL-2; the same strategy has been successfully applied to semaglutide: a once per week GLP-1 agonist for treating diabetes and obesity [30,31]. While simply extending the half-life of the GLP-1 molecule by conjugating a FA molecule in the semaglutide case [32], our approach for the IL-2 molecule actually achieves two goals to greatly enhance the antitumor activity of IL-2: selectively activating CD8^+^ immune cells over Treg cells and extending the in vivo serum half-life and improving the PD profile [32]. 

The selectivity achieved by IL-2K35C-moFA was due to the prevention of IL-2K35C-moFA binding to IL-2Rα through the double effect of the K35C mutation and steric hindrance of FA. Binding to the intermediate affinity IL-2R complex is not impeded, which is demonstrated in the results in vitro and in vivo. The half-life of IL-2WT is so short that twice-daily dosing was required to maintain efficacy in the B16F10 tumor model [10,33]. IL-2K35C-moFA maintained its tumor-suppressive effect when administered once per week, demonstrating the long-term effects of albumin binding by the FA molecule.

Various half-life extended IL-2Rβγc-selective IL-2 variants that utilize alternative strategies for half-life extension and “not alpha” pharmacology have been reported [8,10,11,29]. PEGylation is a widely used strategy for half-life extension and decreasing immunogenicity [34,35,36]; however, in certain cases, the hypothetical impact on long-term safety is still under discussion [37,38,39]. PEGylated IL-2, such as Thor-707, has been shown to reduce the activity of CD8^+^ T cells and NK cells due to the interference of PEG in IL-2Rβ binding. The quality/purity and the cost of the PEG molecule also complicate the CMC of the final product. Thus, the development of alternatives to PEGylation remains crucial [40,41]. IL-2K35C-moFA was engineered to reduce the potential for an immunogenic response, because the FA attachment was positioned at the new mutation site. FA conjugation appears to be much safer and more effective in half-life extension by binding to HSA than the widely adopted PEG strategy. 

## 5. Conclusions

Overall, IL-2K35C-moFA is a FA-conjugated protein with the potential to selectively stimulate CD8^+^ T cells and NK cells. Compared to IL-2WT, IL-2K35C-moFA showed a specifically reduced potency for the stimulation of Treg cells. The combination of selective lymphocyte expansion and its long half-life means IL-2K35C-moFA should be evaluated as a potential human immunotherapeutic in the future.

## 6. Patents

The patent related to FA-conjugated IL-2 molecules presented herein has been filed to the China National Intellectual Property Administration (application number: 202210056065.2).

## Figures and Tables

**Figure 1 cancers-14-04742-f001:**
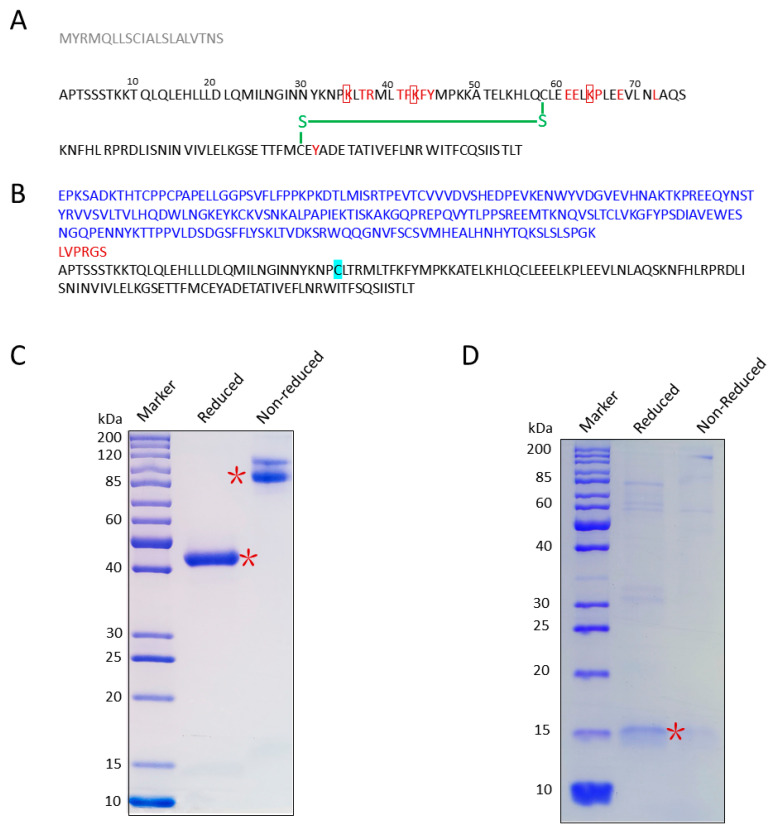
Cloning, expression, and purification of IL-2K35C. (**A**) Amino acid sequence of human IL-2WT. Sequence in grey represents signal peptide. Amino acids in red represent key residues for binding to IL-2Rα. Amino acids enclosed in red squares represent residues mutated to cysteine. The green part represents intramolecular disulfide bond. (**B**) Amino acid sequence of Fc-IL-2K35C. The sequence in blue, red, and black represent Fc, thrombin, and IL-K35C, respectively; the light blue highlighted C represents the mutation site. (**C**) SDS-PAGE patterns of full-length Fc-IL-2K35C. The right panel represents the nonreduced condition.; the left panel represents the reduced condition. * represents the main protein bands. (**D**) SDS-PAGE result of IL-2K35C protein after thrombin digestion. For original SDS-PAGE, see Appendix A.

**Figure 2 cancers-14-04742-f002:**
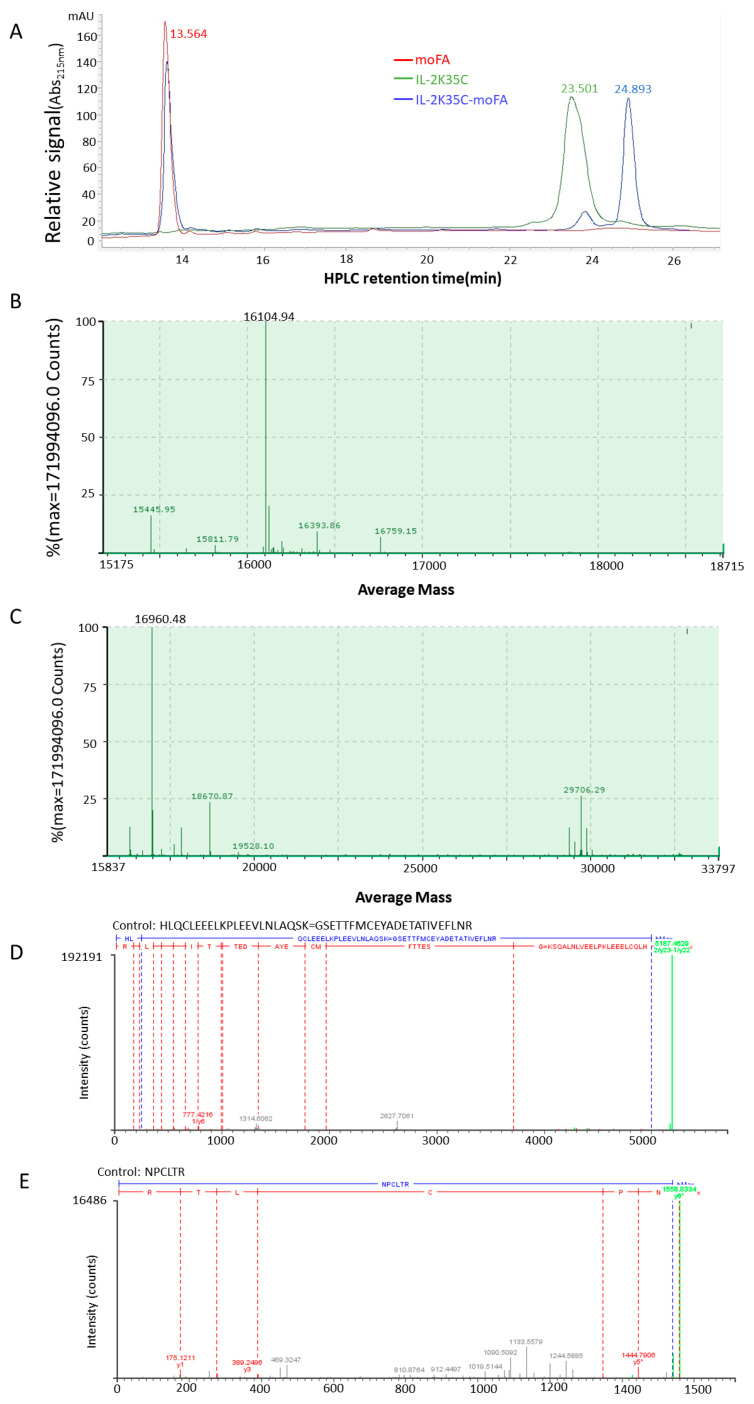
Conjugation and determination of the conjugation site. (**A**) RP-HPLC absorbance profile at 215 nm for the separation of moFA, IL-2K35C, and IL-2K35C-moFA. (**B**) Deconvolution mass results of reduced IL-2K35C. (**C**) Deconvolution mass results of IL-2K35C-moFA. (**D**) LC-MS/MS analysis of trypsin-cleaved peptides and the conjugation site (**E**).

**Figure 3 cancers-14-04742-f003:**
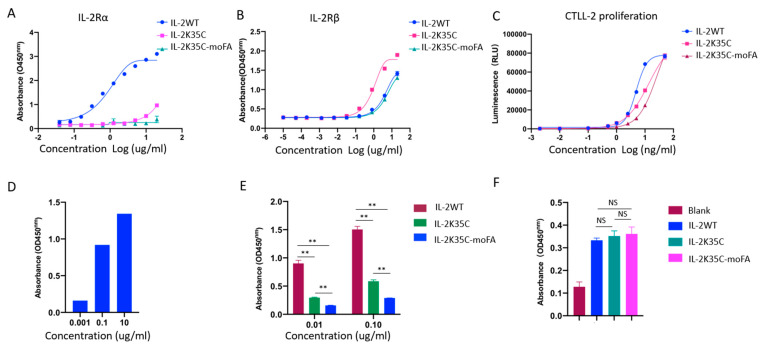
Identification and characterization of IL-2K35C-moFA “not alpha” pharmacology in vitro. (**A**) Binding of IL-2-K35C-moFA to IL-2Rα was determined by ELISA. (**B**) Binding of IL-2-K35C-moFA to IL-2Rβ was determined by ELISA. (**C**) CTLL-2 proliferation in response to IL-2WT, IL-2K35C, and IL-2K35C-moFA. (**D**) STAT5 phosphorylation in a dose-dependent manner in Treg cells mediated by IL-2WT. (**E**) STAT5 phosphorylation in response to IL-2WT, IL-2K35C, and IL-2K35C-moFA in Tregs and in CD8^+^ T cells (**F**). Bars represent an average from 3 replicates. ** *p* < 0.01; NS: not significant.

**Figure 4 cancers-14-04742-f004:**
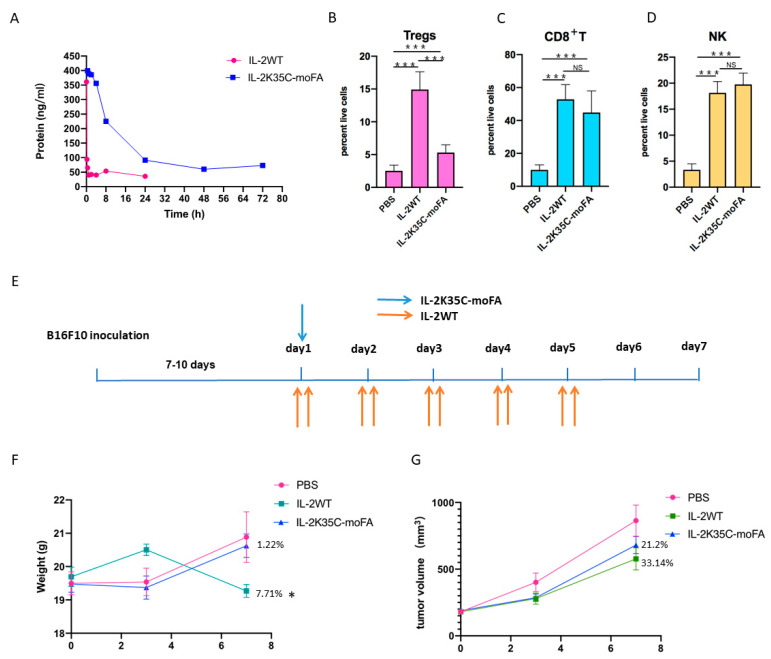
Identification and characterization of IL-2K35C-moFA half-life and pharmacology in vivo. (**A**) Serum levels of IL-2 following intravenous injection of IL-2K35C-moFA and IL-2WT. (**B**) IL-2K35C-moFA induced a weaker stimulation of Tregs relative to IL-2WT. (**C**,**D**) IL-2K35C-moFA induced comparable extent expansions of CD8^+^ T cells and NK cells relative to IL-2WT. *** *p* < 0.001; NS: not significant. (**E**) Timeline diagrams for B16F10 study. IL-2K35C-moFA was administrated subcutaneously on day 1 (blue arrow); IL-2WT was administrated subcutaneously twice daily, indicated by orange arrows. (**F**) Body weight changes; * *p* < 0.05. (**G**) Mean tumor size (±SEM) in mm^3^ was plotted; percentages indicate the percent reduction in mean tumor volume observed on day 7.

**Table 1 cancers-14-04742-t001:** Overlap PCR primer sets.

No.	Oligo
1	CTCTCCCTGTCTCCGGGTAAACTGGTGCCACGCGGTTCGCCTACTTCAAGTTCTACAAAGAAAACACAGCTACAACTGGAGCATT
2	TGGTGAGACAGGGATTCTTGTAATTATTAATTCCATTCAAAATCATCTGTAAATCCAGCAGTAAATGCTCCAGTTGTAGCT
3	AGAATCCCTGTCTCACCAGGATGCTCACATTTAAGTTTTACATGCCCAAGAAGGCCACAGAACTG
4	GCACTTCCTCCAGAGGTTTGAGTTCTTCTTCTAGACACTGAAGATGTTTCAGTTCTGTGGCCTTCT
5	ACCTCTGGAGGAAGTGCTAAATTTAGCTCAAAGCAAAAACTTTCACTTAAGACCCAGGGACTTAATCAGC
6	CATCAGCATATTCACACATGAATGTTGTTTCAGATCCCTTTAGTTCCAGAACTATTACGTTGATATTGCTGATTAAGTCCCTGGG
7	TCATGTGTGAATATGCTGATGAGACAGCAACCATTGTAGAATTTCTGAACAGATGGATTACCTTTTCTCAAAGCATCATCTCAAC
8	TTTGTAATCCAGAGGTTGATTGTCGACTCTAGAATCATCAAGTCAGTGTTGAGATGATGCTTTGAGA

## Data Availability

The data presented in this study is available on request from the corresponding author.

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
