# Peer review of "IL-2K35C-moFA, a Long-Acting Engineered Cytokine with Decreased Interleukin 2 Receptor α Binding, Improved the Cellular Selectivity Profile and Antitumor Efficacy in a Mouse Tumor Model"

_cancers, 2022, doi:10.3390/cancers14194742_

Round 1

Reviewer 1 Report

In the manuscript entitled “IL-2k35c-moFA, a long-acting engineered cytokine with decreased interleukin-2 receptor a binding, improved cellular selectivity profile, and antitumor efficacy in mouse tumor model”, the authors conjugated IL-2K35C with fatty acid  by using maleimide chemistry. They also showed that this conjugated IL-2 variant had longer half-life than wild type IL-2. In addition, in a mouse melanoma model (B16F10), IL-2K35C-moFA showed anti-tumor efficacy as a single dose. However, the novelty is not high enough for publication. There are some concerns which need to be addressed.

1.     The manuscript had some grammar errors which should be corrected. For example, lines 217-218, After conjugation, the protein mass of the main product was 16960.48 Da, which increased to 855.54 Da. This sentence should be changed to “After conjugation, the protein mass of the main product was 16960.48 Da, which increased by 855.54 Da”.

2.     In Figure 4, the authors should perform statistics analysis and show how many times the experiments were done.

3.     B16F10 melanoma is a poorly immunogenic tumor which is sensitive to NK cells, but not CD8+ T cells. Thus, to test the function of IL-2K35C-moFA, another murine tumor model should be established and applied.

Round 2

Reviewer 1 Report

The quality of the current version warrant publication in this journal.